# Cost-Effectiveness Analysis of the Use of Point-of-Care C-Reactive Protein Testing to Reduce Antibiotic Prescribing in Primary Care

**DOI:** 10.3390/antibiotics7040106

**Published:** 2018-12-07

**Authors:** Emily A. F. Holmes, Sharman D. Harris, Alison Hughes, Noel Craine, Dyfrig A. Hughes

**Affiliations:** 1Centre for Health Economics and Medicines Evaluation (CHEME), Bangor University, Normal Site, Bangor, Gwynedd LL57 2PZ, UK; d.a.hughes@bangor.ac.uk; 2Department of Blood Sciences, Betsi Cadwaladr University Health Board, Ysbyty Gwynedd LL57 2PW, UK; Sharman.D.Harris@wales.nhs.uk; 3Head of Pharmacy Primary Care and Community West, Betsi Cadwaladr University Health Board, Pharmacy Office, Eryldon, Campbell Rd, Caernarfon, Gwynedd LL55 1HU, UK, Alison.Hughes9@wales.nhs.uk; 4Public Health Wales, Microbiology Department, Ysbyty Gwynedd, Bangor, Gwynedd LL57 2PW, UK; Noel.Craine@wales.nhs.uk

**Keywords:** economic evaluation, cost–utility analysis, cost-effectiveness analysis, antibiotics, primary care, respiratory tract infection, point-of-care testing, C-reactive protein, antimicrobial resistance, *Clostridium difficile*

## Abstract

More appropriate and measured use of antibiotics may be achieved using point-of-care (POC) C-reactive protein (CRP) testing, but there is limited evidence of cost-effectiveness in routine practice. A decision analytic model was developed to estimate the cost-effectiveness of testing, compared with standard care, in adults presenting in primary care with symptoms of acute respiratory tract infection (ARTI). Analyses considered (1) pragmatic use of testing, reflective of routine clinical practice, and (2) testing according to clinical guidelines. Threshold and scenario analysis were performed to identify cost-effective scenarios. In patients with symptoms of ARTI and based on routine practice, the incremental cost-effectiveness ratios of CRP testing were £19,705 per quality-adjusted-life-year (QALY) gained and £16.07 per antibiotic prescription avoided. Following clinical guideline, CRP testing in patients with lower respiratory tract infections (LRTIs) cost £4390 per QALY gained and £9.31 per antibiotic prescription avoided. At a threshold of £20,000 per QALY, the probabilities of POC CRP testing being cost-effective were 0.49 (ARTI) and 0.84 (LRTI). POC CRP testing as implemented in routine practice is appreciably less cost-effective than when adhering to clinical guidelines. The implications for antibiotic resistance and *Clostridium difficile* infection warrant further investigation.

## 1. Introduction

Respiratory tract infections (RTI) are the most common presenting complaint in primary care and the most common reason for antibiotic prescribing in Europe [1]. Whilst antibiotics will benefit lower respiratory tract infections (LRTI) of bacterial origin, they are often prescribed inappropriately, such as for viral upper RTI, putting patients at risk of adverse effects with limited or no therapeutic benefit [2]. Identifying patients presenting with bacterial LRTI that require an antibiotic represents a challenge to healthcare professionals. Unnecessary antibiotic prescribing also increases the risk of development of antimicrobial resistance [2,3,4,5] and *Clostridium difficile* infection [6]. The vital importance of addressing antimicrobial resistance at a local, national and international level is widely recognized [7]. O’Neill’s report for the United Kingdom (U.K.) government [8], the National Institute of Health and Care Excellence (NICE) guidelines on antimicrobial stewardship [9], and the Welsh Government delivery plan for Wales “Tackling antimicrobial resistance and improving antibiotic prescribing” [10], all highlight the urgency of the task. 

To avoid unnecessary use of antibiotics, the NICE advises considering point-of care (POC) C-reactive protein (CRP) testing in primary care in patients presenting with symptoms of LRTI, when clinical assessment is not conclusive and it is not clear whether antibiotics should be prescribed [11]. The clinical guideline recommends that antibiotic therapy should not be routinely offered if the CRP is <20 mg/L; a delayed antibiotic prescription should be considered if the CRP is 20–100 mg/L; and antibiotic therapy should be offered if the CRP is >100 mg/L [11]. 

In Wales, the national policy is for wider use of POC CRP testing as a prognostic tool in primary care to aid clinical decisions about the appropriateness of antibiotic prescribing [12]. Implementation of POC CRP testing across Wales requires a clear understanding of the potential costs and benefits associated with its use. A meta-analysis including 10,005 patients showed that CRP testing was associated with a significant reduction in antibiotic prescribing at index consultation but not at 28-day follow-up, and did not impact on patient satisfaction [13]. A more recent review [14] suggested that reductions in antibiotic prescribing attributable to POC CRP testing range from 23% to 36% [15,16,17,18,19]. Economic evaluations of POC CRP testing in managing RTIs suggest potential cost savings [20,21,22,23]; however, these are of variable quality and based on key assumptions concerning the effectiveness of implementation of POC CRP testing, laboratory support costs, and connectivity to wider healthcare systems [24]. A formal analysis of the cost-effectiveness of CRP testing in patients presenting with acute respiratory tract infection (ARTI) in the context of routine National Health Service (NHS) primary care service delivery, which considers the full costs of implementation, is therefore required. Given the objective of POC CRP testing is to reduce unnecessary antibiotic prescribing, in order to conserve the effectiveness of current antimicrobials, economic evaluations of POC CRP testing have the added challenge of how best to capture the cost of antimicrobial resistance [25]. 

The aim of the economic evaluation was to estimate the cost-effectiveness of POC CRP testing of adult patients presenting with symptoms of ARTI in routine use, acknowledging widespread non-compliance with clinical guidelines, and to compare this with the cost-effectiveness of testing according to clinical protocol. The study design is an economic model based on empirical data from a published study [26]. This is the first study, to our knowledge, to consider the influence on the cost-effectiveness of POC CRP testing of deviations from clinical guidelines—such as prescribing antibiotics regardless of test result—that are common occurrences in routine practice. Evidence generated by this evaluation suggests pragmatic use of POC CRP testing is considerably less cost-effective than when adhering to clinical guidelines, and that including the cost of antimicrobial resistance in the model improves the cost-effectiveness of POC CRP testing.

## 2. Results

### 2.1. Base Case Analysis

The study population had a median age of 48.5 years, and 69% were female (one participant out of 71 was lost to follow-up and was excluded from the analysis) [26]. The mean number of CRP tests received over 28 days was 1.03 per patient, at a cost of £9.85 (95% confidence interval [CI] 9.63 to 10.42). With POC CRP testing 18/70 patients received antibiotics (Table 1). At least 10 of these prescriptions would be considered unnecessary according to CRP testing guidelines (143/1000 prescriptions). Compared with standard care, the modelled incremental cost of the POC CRP strategy in the ARTI population, was £11.92 (95% CI 9.35 to 15.39); the main cost drivers were the costs of the test and re-consultation. Modelled differences in quality-adjusted-life-years (QALYs) between standard care and POC CRP were 0.0006 (95% CI −0.0006 to 0.0019), which is equivalent to approximately five additional hours of perfect health over 28 days. The base-case incremental cost-effectiveness ratio (ICER) is therefore £19,705 per QALY gained. The cost-effectiveness of POC CRP testing for ARTI is estimated to be £16.07 per antibiotic prescription avoided (£11.25 per 1% reduction in antibiotic prescribing). 

### 2.2. Sensitivity and Scenario Analyes

#### 2.2.1. One-Way Sensitivity and Threshold Analyses

The threshold analysis revealed that if each POC CRP test were 18 pence more expensive, the ICER would exceed £20,000 per QALY gained (Table 2). If CRP test usage were to fall by 5% (from 37 to 35 tests/1000 patient-years) the ICER for ARTI will exceed £20,000 per QALY. Adjusting the proportion of patients seen by a general practitioner (GP) or independent nurse prescriber (INP) consultation varied the ICER from £16,288 to £19,749 per QALY. 

#### 2.2.2. Probabilistic Sensitivity Analysis

The cost-effectiveness plane for the ARTI base case analysis is illustrated in Figure 1. The distribution of the simulations indicates that POC CRP testing results in higher utility (health gain) but at higher cost in 75% of simulations. The corresponding cost-effectiveness acceptability curve (CEAC) indicates the probabilities of POC CRP testing for ARTI being cost-effective were 0.49 and 0.63 at the £20,000 and £30,000 per QALY thresholds, respectively (Figure 2). 

#### 2.2.3. Scenario Analyses

The scenario of restricting CRP testing to patients with symptoms of LRTI for >12 hours; is represented by a sub-group of 20 patients with a median age 48.5 years, 55% female. Based on the observed CRP test results for the study population (no CRP >100 mg/L) [26], the model predicts a 100% reduction in antibiotic prescribing at an incremental cost of £9.31 per patient (95% CI 7.24 to 14.11) (Table 1). The increase in total cost is attributable to the cost of POC CRP testing (£10.05 per patient), which is not outweighed by the savings in antibiotic prescribing costs (−£2.53 per patient). The cost–utility analysis indicates that POC CRP testing for LRTI, according to protocol, is associated with a 0.0021 QALY gain (95% CI −0.0011 to 0.0058), equivalent to about 19 quality-adjusted hours, with a resulting ICER of £4390 per QALY gained. There is capacity for test prices to increase 4-fold before the ICER reaches the £20,000 per QALY threshold. Probabilistic sensitivity analysis for this scenario indicates that POC CRP testing is associated with increased QALYs and higher cost in 88% of simulations and a probability of being cost-effective at a threshold of £20,000 per QALY threshold, of 0.84. 

Additional scenarios indicate that the ICER decreases (i) under the assumption of equal re-consultation rates for both standard care and CRP testing, regardless of antibiotic prescribing; (ii,a&b) when accounting for antimicrobial resistance; (iii) when a 7-day course of amoxicillin is dispensed by a doctor; and (vii) if the machine-life is extended to 10-years. POC CRP testing is dominant when (ii,c) the cost of each antibiotic prescription includes the global cost of treating antimicrobial resistance [27]. POC CRP testing is less cost-effective (higher ICER) when the model includes (iv) hospitalizations related to ARTI; or (v) a lower rate of antibiotic prescribing in standard care (53%) reduces the cost-effectiveness (increases the ICER). Testing is no longer cost-effective if (vi) prescribing guidelines are adhered to (5-day instead of 7-day course of amoxicillin). 

## 3. Discussion

### 3.1. Key Findings 

The model suggests that as implemented in routine primary care (for all adults with symptoms of ARTI for >12 hours where the antibiotic decision unclear) POC CRP testing is borderline cost-effective [28]. There are a number plausible scenarios where testing outside the recommendations of the NICE clinical guideline exceeds the cost-effectiveness threshold. Closer adherence to the NICE CRP recommendation, however, by restricting testing to adults with symptoms of LRTI, and prescribing appropriate courses of antibiotics, results in a more favourable ICER. The main cost driver is the cost of a CRP test which, for ARTI must be below £9.76 per test to be considered cost-effective. This represents a small increase (+£0.18) from the current estimate, that may easily be caused by changes in unit costs or throughput (such as a 5% increase in agent costs, £6/month increase in laboratory support costs, 17% increase in equipment costs, or 5% reduction in the use of testing). Including the cost of antimicrobial resistance in the model improves the cost-effectiveness of POC CRP testing, but there are uncertainties associated with specifying this cost, and resistance would also affect the efficacy of the antibiotic, which was not explicitly considered in this analysis. This evaluation did, however, identify a 74% absolute reduction in antibiotic prescribing (from 70 to 18 prescriptions) and an 89% reduction in unnecessary prescribing for adults with ARTI. It is plausible that reductions in unnecessary prescribing could conserve the effectiveness of current antimicrobials. 

### 3.2. Comparisons to Other Studies

We are aware of four published economic evaluations of POC CRP testing for RTI. Cals et al. (2011) conducted a cost-effectiveness analysis of POC CRP testing by GP versus standard care based on data from a cluster randomised factorial clinical trial of 431 patients with LRTIs recruited in 40 GP practices [20]. They reported an ICER of €5.79 per 1% reduction in antibiotic prescribing (corresponding to £4.02). Oppong et al., (2013) conducted an economic evaluation based on an observational study of the presentation, management, and outcomes of patients with acute cough and LRTI in primary care settings in Norway and Sweden, and reported an ICER of €9,391 per QALY gained [23]. NICE based their analysis on the incremental QALY (0.0012) reported by Oppong et al. (2013) [23] to estimate an ICER of £15,763 per QALY gained [11]. This included the cost of hospitalization, as evaluated in our scenario analysis. More recently, Hunter (2015) conducted an analysis which included both hospitalizations and the costs of complications of antibiotic prescribing, and reported CRP testing dominates current practice [22]. 

Studies of POC CRP testing are characterized by uncertainty that may be attributable to valuation of the test, heterogeneity in study population, and the subjectivity of the indication. Regents were 75% of costs in the analysis by Cals et al. (2011) [20], but accounted for only 37% the current analysis. The proportion of cost attributable to reagent was similar in the analysis by Hunter (2015) [22], however, costs were limited to reagent, depreciation and staff time. Other studies provide no details of what was included in the valuation [11,23]. Furthermore, defining the eligible patient population requires a subjective judgement, i.e. “where the antibiotic decision is unclear”, that represents a challenge for economic modelling. By considering the use of testing as routinely implemented in practice, our analysis reflects the judgements of GPs. 

### 3.3. Strengths

To our knowledge, this is the first study in the UK to model the cost-effectiveness of POC CRP testing using data from use in routine clinical practice, and to include support costs for the management of POC testing in primary care. The probabilities in the model were based on individual patient-level data, undertaken over 3 months, in a GP surgery that had a high rate of antibiotic prescribing. The analysis, therefore, reflected the real world situation of protocol deviations and provides an estimate of the cost-*effectiveness* (as opposed to the cost-efficacy) of testing. Unlike previous studies, our analyses also attempted to incorporate the long-term cost implications of antimicrobial resistance.

### 3.4. Limitations

The time horizon of the model is unable to assess the longer-term effects of unnecessary antibiotic prescribing, such as the increased antibiotic resistance and increased risk of *Clostridium difficile* infection [5,29]. The scenario analysis of antibiotic resistance is limited to a projected cost and is not reflective of patients’ health-related quality of life. The utilities associated with RTI are baseline estimates and the model assumes this is constant for the duration of the illness; due to a lack of disaggregated longitudinal data. The disutility of common adverse events, such as diarrhoea, are assumed to be captured by time to full recovery, which may underestimate resource use and overestimate utility associated with antibiotic prescribing. Finally, this model is representative of outlying practice. The estimate of 100% prescribing of antibiotics in standard care, and non-compliance to POC CRP guidelines, is less likely in other places. 

### 3.5. Implications

This analysis highlights the reduction in cost-effectiveness attributable to protocol deviation, as is expected in a routine clinical setting. The cost-effectiveness of the POC CRP testing strategy is highly sensitive to the cost of the test, therefore when interpreting the result in other settings; consideration should be given to test cost drivers, such as machine throughput. Economies of scale are likely to be limited by the size of the practice population and uptake of the test by healthcare professionals. 

### 3.6. Future Research Directions

Further research is required to capture the effects of antimicrobial resistance and increased risk of *Clostridium difficile* infection, both of which are associated with increased NHS costs [27,29], morbidity and mortality; and have long term impacts on health-related quality of life and society [25,29]. The appropriateness of QALYs for use in evaluation of acute conditions, such as respiratory tract infections, warrants further exploration; alternative methods such as willingness to pay have been suggested [30]. Beyond the testing strategy, research is also required on behavioural aspects, such as protocol adherence by healthcare professionals and the influence on the test on medication adherence by the patients.

How best to assess the value of any intervention to reduce antimicrobial resistance is a methodological challenge, which requires adequate measurement of the expected rate of growth of antimicrobial resistance and associated outcomes over time [31]. Rothery and colleagues [31] recently outlined a framework for value assessment of new antimicrobials that uses modelling to estimate infection transmission dynamics, associated resistance, and economic outcomes, for alternative treatment strategies. The report highlights several implications for health technology assessment [31]. Modelling infection transmission dynamics and resistance outcomes over time is complex and relies on the multidisciplinary teams (mathematical modellers, epidemiologist, data experts, clinical experts, and health economists). There is greater reliance on observational data and dynamic transition modelling rather than cohort statistic modelling. This is associated with more extensive and systematic use of expert elicitation methods; and, of model calibration for interring values for unobservable parameters or limited efficacy data. Furthermore, there are difficulties in measuring and valuing health—with limited data on utility values, difficulty measuring health related quality of life in short severe infections, and difficulty measuring the disutility and costs of onward transmission of infections to the wider population. Typical health technology appraisal, such as that conducted by NICE, is based on the “average” patient receiving a treatment for a specified indication. When considering the consequences of antibiotic prescribing benefits and costs extend to a wider population, within which diversity in settings (e.g., community care, intensive care) will influence the spread of infections in the population. Given this extension to the population level, Rothery and colleagues, also highlight the issue of an indefinite time horizon [31]. Future research therefore needs to address the challenge of how to formally characterize the value of interventions with the potential to reduce antimicrobial resistance and to measure the opportunity cost used to guide this longer term and more global decision. Rothery and colleagues suggest the following uncertainties need to be considered: prevalence of infections, resistance patterns over time, stock of future antibiotics, lag periods before resistance, irreversible impacts. A broader perspective may also be necessary, to explore issues such as the insurance value of avoiding major health consequences if antimicrobial resistance becomes substantially worse [25], and antibiotic use in farming with associated events, for example, standards relating to the import of diary and meat into the United Kingdom and associated trade deals on departure from the European Union [32]. 

## 4. Materials and Methods 

We conducted a cost-effectiveness analysis of antibiotic prescribing conditional on POC CRP testing for adults presenting in primary care (GP practice) with symptoms of ARTI for >12 hours versus immediate antibiotic prescription (current standard of care). The analysis had a 28-day time horizon, conducted from the perspective of the NHS in the United Kingdom.

### 4.1. Economic Model

A decision analytic model was developed with a time horizon of 28 days from the index consultation (Figure 3). The model was structured to represent the following care pathways [11]: (1) standard care, in which patients receive no test (as per current practice) and instead receive an antibiotic prescription for immediate use, and (2) a strategy of testing for CRP, where patients have three potential prescription outcomes, no antibiotic prescribed, antibiotic prescribed for immediate use, or for delayed use [33]. Patients who are offered a delayed prescription are offered a prescription for use at a later date if symptoms worsen. The model accounts for whether or not a delayed prescription, issued at the index consultation (based on CRP 20–100 mg/L), is dispensed.

The model also takes account of re-consultations occurring within 28 days where patients on the CRP testing pathway receive no antibiotic prescriptions (CRP <20 mg/L). At re-consultation, the model considers whether or not patients receive a repeat CRP test and thereafter, whether antibiotics are prescribed or not. Patients who were not going to receive an antibiotic prescription irrespective of testing did not enter the model.

#### 4.1.1. Clinical Parameters 

The model was parameterized using data from purposive reviews of the literature, in line with standard methodology for populating economic models [34]. These supplemented data from a published study [26] undertaken in a general practice surgery that serves 10,200 patients in Anglesey, North Wales, UK. The practice was in the top one percentile in England and Wales for antibiotic prescribing [35]. A POC CRP analyser was introduced in November 2015 and used for 3 months, based on the NICE clinical guideline for pneumonia [11]. POC testing was supported by a laboratory POC team as specified by national policy [12]. Within the study [26], data were collected on POC CRP test results, antibiotic decision, re-consultation within 28 days for the same complaint, and, the outcome of any decision to prescribe antibiotics at any re-consultation. Dispensing of delayed prescriptions within 28 days of the index consultation was determined by retrospective review of the online NHS Wales Shared Services “primary care prescribing catalogue” [36]. Patients who were lost to follow-up were excluded from the analysis. Clinical parameter estimates (probabilities) are listed in Table 3. 

The standard care arm of the economic model is based on routine clinical practice at the same location. We assumed that 100% of patients (adults with symptoms of ARTI for 12 or more hours where the antibiotic decision is unclear) in the standard care arm of the model received an immediate prescription. Re-consultation rates in the standard care arm were modelled on the rate of re-consultation in the CRP pilot study for patients who received antibiotics at index consultation (0% at 28 days; [26]. 

All patients in the model had a risk of hypersensitivity reaction to antibiotic therapy that is independent of CRP testing strategy or re-consultation rates. As there were no recorded cases of adverse drug reactions in the pilot study, the probability of anaphylaxis was taken as that for amoxicillin, and assumed to be 1 in 10,000 people [37]. 

#### 4.1.2. Resource Use and Costs

Resource use included GP or INP index and re-consultation, CRP testing, antibiotic prescription and treatment of adverse drug reactions. The rate of index consultation and subsequent consultations, and the proportion of patients who consulted with a GP or INP, face-to-face or by telephone, was taken from Hughes et al., (2016) [26] and assumed to be the same for CRP testing as for standard care. Re-consultation with the same complaint occurred in 7/50 cases with upper respiratory tract infection (URTI) and 1/20 cases with LRTI. The cost of a consultation was based on a mean duration of 9.22 minutes with a GP [38] and assumed to be 15 min with an INP (Table 3). Costs of GP and nurse-led telephone triage were based on national figures [39]. 

Antibiotic prescriptions were assumed to be amoxicillin 500-mg capsules three times daily for 7 days, informed by retrospective review of prescribing at the GP practice involved in the study, where there were no records of a 5-day course being prescribed (as recommended). Local prescribing data showed that in this primary care cluster, 22% of amoxicillin prescriptions (all indications) were prescribed for 5 days. Prescription costs were taken from the British National Formulary at the Drug Tariff price [40]. An additional dispensing fee was included for every dispensed prescription, based on the mean dispensing rate per item for community pharmacists at the location [41]. 

The cost of a managing an anaphylaxis reaction (adverse drug reaction) consisted of ambulance treatment and transport, followed by emergency medicine investigation and treatment [41,42]. The model assumed all patients who experience a hypersensitivity reaction survive and are prescribed clarithromycin as an alternative antibiotic. 

The cost of CRP testing included the cost of performing a test using the Alere Afinion AS100 analyser (Alere; MA, United States) (Table 4). This included the fixed cost of purchasing the analyser and variable costs of its use in clinical practice. Variable costs included consumables, internal quality control (IQC), external quality assurance (EQA), maintenance costs, connectivity, and hospital-based laboratory POC testing team assistance. Extra equipment was required to connect the analyser into the All Wales Laboratory Information System (LIMS), with the added cost of a company to connect the analyser so that results can be viewed in the patient record. The cost per test was calculated assuming a machine life of 5 years, based on the manufactures estimate, and the projected annual number of tests based on the number of test performed in 3 months [26]. In the base-case analysis, the test was assumed to be performed by a health care assistant (HCA), in addition to the standard consultation with GP/INP. 

#### 4.1.3. Health State Utilities

Health state utilities were estimated for presenting complaint (Table 3). The utility associated with LRTI was the mean EuroQol EQ-5D-3L value at baseline for patients in Wales (*n* = 181) participating in an observational study of the management of patients with acute cough and LRTI in primary care [1,45]. For URTI, the decrement used in the NICE Clinical Guideline on antibiotic prescribing for respiratory tract infections [46,47], was applied to the U.K. age-specific population norm [44]. The utility of each RTI was assumed constant for the duration of symptoms [18], after which patients returned to the UK age-specific population norm [44]. Duration of symptoms associated with LRTI and URTI was the length of time from index consultation to patient-reported full recovery, from a study of patients with LRTI and rhinosinusitis [18]. Quality-adjusted life-years (QALYs) were then calculated for the 28-day time horizon of the model. 

The model used price year 2016–17 for all costs. Discounting was not required due to short time horizon of the model. 

### 4.2. Analysis

#### 4.2.1. Cost Per QALY

The primary analysis for POC CRP testing for adults with symptoms of ARTI for >12 hours, versus, immediate antibiotic prescription, resulted in the calculation of the incremental cost-effectiveness ratio (ICER), as follows:
$$ICER=\frac{COST_{with\;test}-COST_{standard\;care:no\;test}}{OUTCOME_{with\;test}-OUTCOME_{standard\;care:no\;test}}$$

#### 4.2.2. Cost-Effectiveness Analysis 

A secondary, cost-effectiveness of POC CRP testing considered the cost per antibiotic prescription avoided. The ICER was calculated as the incremental cost divided by the total number of prescriptions avoided (*N* Prescriptions with test, *N* Prescriptions standard care with no test).

#### 4.2.3. Base-Case Analysis

The care pathways in the base-case analysis allowed for deviation from the NICE guideline [11] by including all adult patients (including upper RTI) and reflecting the real world use of CRP testing. This used observational data to reflect actual clinical practice [26] which exhibits variable compliance with clinical and prescribing guidelines. 

#### 4.2.4. Sensitivity Analyses

A threshold analysis was conducted to establish the cost and throughput of testing at which the ICER met the NICE cost-effectiveness threshold of £20,000 per QALY [28]. A one-way sensitivity analysis was conducted on the probability that patients are seen by GP or INP at index and re-consultation. Probabilistic sensitivity analysis was also performed, using Monte Carlo simulation with 10,000 replications sampled from the distributions presented in Table 3. A CEAC was constructed to illustrate the probability of testing being cost-effective at given thresholds of cost-effectiveness [49].

#### 4.2.5. Scenario Analyses 

The NICE guidance advises that POC CRP testing should only be used for patients presenting with symptoms of LTRI for >12 hours, where the antibiotic decision is unclear. A scenario analysis was therefore conducted, assuming that all patients are treated according to the NICE (2014) guideline [11], and that if antibiotics were required, they were prescribed amoxicillin 500 mg three times daily for 5 days (BCUHB Adult Antimicrobial Guide). The model structure is displayed in Figure 4.

Alternative modelled scenarios considered: (1) the possibility that patients would re-consult in standard care at the same rate as following POC CPR testing, irrespective of antibiotic outcome; (2) the impact of inappropriate prescribing on antibiotic resistance, based on costs extracted from Oppong et al., (2016) [27]; (3) antibiotics being dispensed by doctors rather than by community pharmacists, as might happen in more rural settings, and using local rates (personal communication); (4) the impact of hospitalizations related to ARTI on the cost-effectiveness of testing, based on published probabilities for CRP (0.009) and standard care (0.003) [11]; (5) the probability of reduced antibiotic prescribing in standard care (to 53%) [11]; and (6) the prescribing of 5 days’ supply of amoxicillin for ARTI (as per guideline); and (7) machine-life extended to 10-years. All parameter values used in the scenario analyses are detailed in table 5. The three estimates for the cost of antibiotic resistance were based on the annual cost of resistance in the United States (U.S.) ($55 billion) [50], the cost of multidrug resistance in the European Union (EU) (1.5 billion euro) [51], and the cost of global resistance over a 35-year period ($2.8 trillion annually) [8]. Oppong and colleagues [27] estimated the cost per prescription in each scenario by calculating the cost of annual cost of resistance, divided by the annual number of prescriptions in each region—assuming antibiotic prescribing is the main cause of resistance. In the current analysis, estimates were converted into pounds sterling and then inflated to price year 2016–2017. 

All analyses were performed in Microsoft Excel 2010 (Microsoft Corp., Redmond, WA, USA) and the study is reported according to the Consolidated Health Economic Evaluation Reporting Standards [52] [Appendix A].

## 5. Conclusions

POC CRP testing for adults where the antibiotic decision is unclear, is borderline cost-effective, however the results are favourable when restricted to patients with LRTI symptoms only adhering to protocol. Modelling pragmatic use of testing, reflective of practice, using observed data that deviated from protocols for both CRP-guided prescribing, and the length of prescription thereafter; illustrated the potential for variation in cost-effectiveness in clinical practice. POC testing for patients with upper respiratory tract infection is less likely to be cost-effective and comes with a higher opportunity cost for alternative use of NHS resources. The results of this economic evaluation are subject to considerable uncertainty, and therefore further empirical research is necessary. 

## Figures and Tables

**Figure 1 antibiotics-07-00106-f001:**
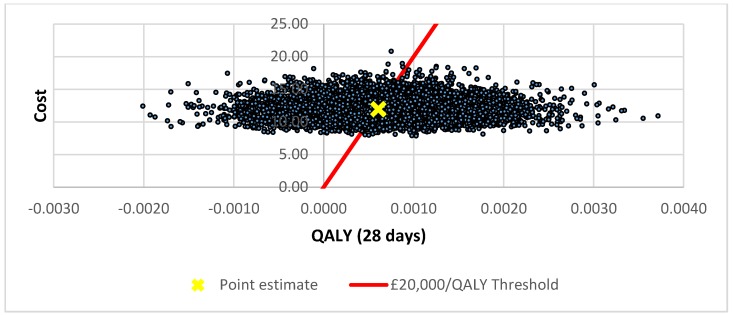
Cost-effectiveness plane for pragmatic use of POC CRP testing.

**Figure 2 antibiotics-07-00106-f002:**
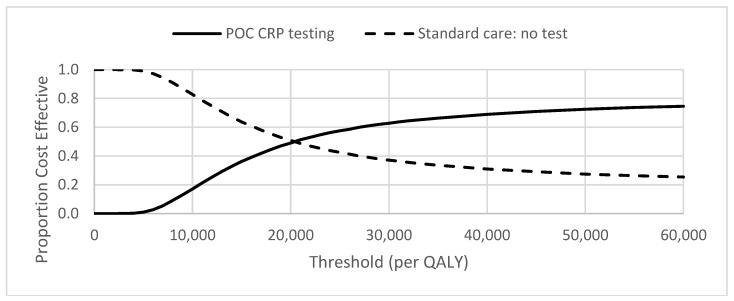
Cost-effectiveness acceptability curve (CEAC) for pragmatic use of POC CRP testing.

**Figure 3 antibiotics-07-00106-f003:**
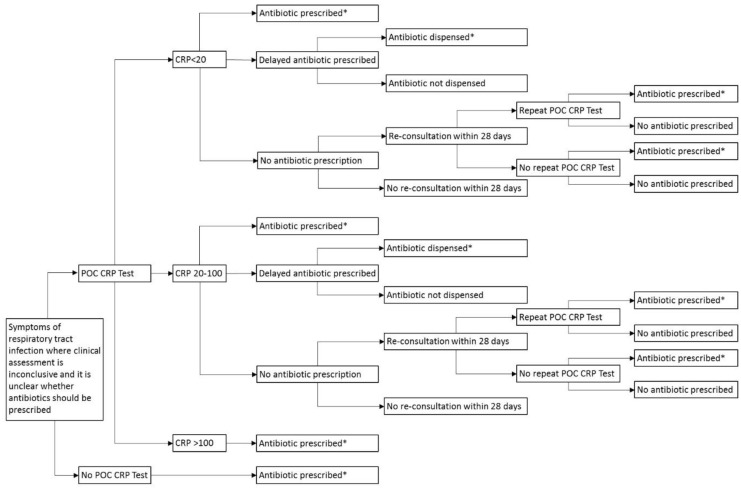
Decision tree for pragmatic use of POC CRP testing, reflective of practice. POC: Point-of-care; CRP: C-reactive protein; *Amoxicillin 500 mg three times daily for 7 days.

**Figure 4 antibiotics-07-00106-f004:**
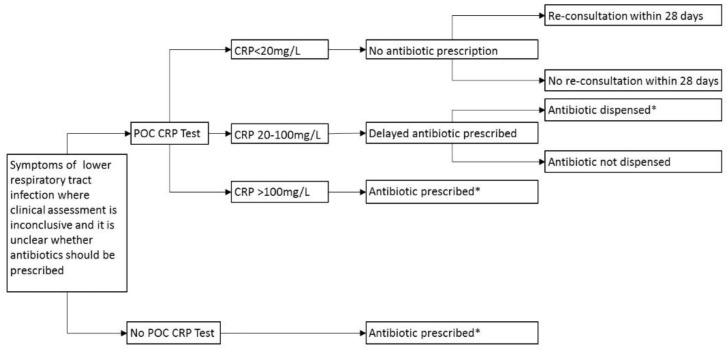
Decision tree for POC CRP testing adhering to guidelines, LRTI only. POC: Point-of-care; CRP: C-reactive protein; * Amoxicillin 500 mg three times daily for 5 days.

**Table 1 antibiotics-07-00106-t001:** Cost-effectiveness of point-of-care (POC) C-reactive protein (CRP) testing for adults with symptoms of acute respiratory tract infection (ARTI) for >12 h where the antibiotic decision is unclear versus immediate antibiotic prescription. CI: Confidence Interval; ICER: Incremental cost-effectiveness ratio.

**Results for 28 Days and 1 Patient**	**Intervention**			**Control**			**Increment**			**ICER**
**Pragmatic Use of Testing, Reflective of Practice**	**Mean**	**95% CI**	**95% CI**	**Mean**	**95% CI**	**95% CI**	**Mean**	**95% CI**	**95% CI**	
Costs (£, over 28 days)										
Consultation cost	41.75	39.36	44.71	37.46	37.39	37.54	4.28	1.91	7.26	
CRP testing	9.85	9.63	10.42	0.00	0.00	0.00	9.85	9.63	10.42	
Antibiotic prescription	0.74	0.47	1.04	2.89	2.85	2.89	−2.15	−2.41	−1.84	
Adverse drug reaction to antibiotic	0.02	0.01	0.02	0.06	0.05	0.07	−0.04	−0.06	−0.03	
Total cost	52.35	49.76	55.79	40.41	40.32	40.48	11.94	9.35	15.39	
Effectiveness (over 28 days)										
Antibiotic prescription avoided (*N*)	0.74	0.64	0.84	0.00	0.00	0.01	0.74	0.64	0.84	
Cost-effectiveness (over 28 days)										
£/prescription (Rx) avoided							£16.07/Rx avoided			
Utility (for 28 days)										
Quality-adjusted-life-year (QALY)	0.0615	0.0512	0.0706	0.0609	0.0507	0.0700	0.0006	−0.0006	0.0019	
Cost–utility										
£/QALY							£19,705/QALY			
Probabilistic result	%									
Probability cost-effective at £20,000/QALY	49.06									
Probability cost-effective at £30,000/QALY	62.82									
**Results for 28 days and 1 patient**	**Intervention**			**Control**			**Increment**			**ICER**
**Adhering to guidelines**	**Mean**	**95% CI**	**95% CI**	**Mean**	**95% CI**	**95% CI**	**Mean**	**95% CI**	**95% CI**	
Costs (£, over 28 days)										
Consultation cost	38.73	36.90	42.76	36.89	36.70	37.06	1.84	0.05	5.85	
CRP testing	10.05	9.58	10.68	0.00	0.00	0.00	10.05	9.58	10.68	
Antibiotic prescription	0.00	0.00	0.00	2.53	2.42	2.53	−2.53	−2.47	−1.86	
Adverse drug reaction to antibiotic prescription	0.00	0.00	0.00	0.06	0.05	0.07	−0.06	−0.07	−0.04	
Total cost	48.79	46.66	53.53	39.48	39.25	39.62	9.31	7.24	14.11	
Effectiveness (over 28 days)										
Antibiotic prescription avoided (*N*)	1.00	0.75	0.98	0.00	0.00	0.04	1.00	0.73	0.98	
Cost-effectiveness (over 28 days)										
£/prescription (Rx) avoided							£9.31/Rx avoided			
Utility (for 28 days)										
Quality-adjusted-life-year (QALY)	0.0577	0.0536	0.0612	0.0556	0.0509	0.0594	0.0021	−0.0011	0.0058	
Cost–utility										
£/QALY							£4,390/QALY			
Probabilistic result	%									
Probability cost-effective at £20,000/QALY	84.10									
Probability cost-effective at £30,000/QALY	86.33									

**Table 2 antibiotics-07-00106-t002:** Results of sensitivity and scenario analyses.

Analysis	Parameter Description	Pragmatic Analysis Reflective of Practice: Acute Respiratory Tract Infection (ARTI)	Adhering to Protocol: Lower Respiratory Tract Infection (LRTI) only
		**Incremental Cost-Effectiveness Ration (ICER)**	**Incremental Cost-Effectiveness Ration (ICER)**
Base case		£19,705	£4390
Threshold analysis: cost of test			
	Cost of test = £0	£3449	DOMINANT
	Cost of test = +£0.18 (£9.76) ~2% increase	£20,010	n/a
	Cost of test = +£31.60 (£41.18) ~4-fold increase	n/a	£20,036
Threshold analysis: Scale of testing: number of tests per year (base case = 376)			
	Acute respiratory tract infection (ARTI) only (*n* = 280 test per practice per annum)	£21,834	n/a
	Lower respiratory tract infection (LRTI) only (*n* = 80 test per practice per annum)	n/a	£11,094
	5% decrease (358 tests per practice per year)	£20,017	n/a
	90% decrease (39 tests per practice per year)	n/a	£20,046
One-way sensitivity analysis: Healthcare professional at index and re-consultation			
	General practitioner (GP): Independent Nurse Prescriber (INP) 50:50	£18,081	£4193
	GP	£19,749	£4410
	Nurse	£16,288	£3976
Scenario analyses			
i	Re-consultation rate		
	Equal in each arm i.e. standard care = Point-of-care (POC) C-reactive protein (CRP) pilot study	£12,638	£3520
ii	Cost of antimicrobial resistance per prescription over 28 days		
	(a) European	£19,525	£4321
	(b) U.S.	£13,854	£2140
	(c) Global	Dominant	Dominant
iii	Dispensing item fee at local dispensing doctor rate £1.90	£19,361	£4258
iv	Hospital admission	£26,927	£ 6454
v	Antibiotic prescribing in standard care 53%	£20,277	£4533
vi	Amoxicillin prescription		
	500 mg capsules three times daily for 5 days	£20,146	n/a
	500 mg capsules three times daily for 7 days	n/a	£4220
vii	CRP analyser machine life 10-years	£19,183	£4238

**Table 3 antibiotics-07-00106-t003:** Model input parameters: probabilities, costs and utilities.

Parameter	Point Estimate	Distribution ^1^	References
**Probabilities**			
Antibiotics at index | C-reactive protein (CRP) > 100 mg/L	1.00	Fixed	[11]
Antibiotics at index consultation | no CRP	1.00	Fixed	Assumption ^2^
Anaphylactic reaction to antibiotic prescription	0.0001	Beta (1, 10, 000)	[37]
**ARTI Observed Data**			
CRP < 20 mg/L	0.77	Dirichlet (54, 16, 0)	[26]
CRP 20–100 mg/L	0.23	Dirichlet (16, 54, 0)	[26]
No antibiotics at index consultation | CRP < 20 mg/L	0.93	Dirichlet (50, 2, 2)	[26]
Delayed prescription at index consultation | CRP < 20 mg/L	0.04	Dirichlet (2, 50, 2)	[26]
Delayed prescription at index consultation not dispensed | CRP < 20 mg/L	1.00	Beta (1, 0)	[26]
Antibiotics at index consultation | CRP < 20 mg/L	0.04	Dirichlet (2, 50, 2)	[26]
No antibiotics at index consultation | CRP 20–100 mg/L	0.38	Dirichlet (6, 10, 0)	[26]
Antibiotics at index consultation | CRP 20–100 mg/L	0.63	Dirichlet (10, 6, 0)	[26]
No re-consultation within 28 days | CRP < 20 mg/L	0.86	Beta (43, 7)	[26]
No re-consultation within 28 days | CRP 20–100 mg/L	0.83	Beta (5, 1)	[26]
No repeat CRP at re-consultation| CRP < 20 mg/L	0.71	Beta (5, 2)	[26]
No repeat CRP at re-consultation| CRP 20–100 mg/L	1.00	Beta (1, 0)	[26]
CRP guided no antibiotic decision at re-consultation | CRP <20 mg/L	1.00	Beta (2, 0)	[26]
Antibiotics at re-consultation | CRP < 20 mg/L at index, CRP not repeated at re-consultation	1.00	Beta (5, 0)	[26]
Antibiotics at re-consultation | CRP 20–100 mg/L at index, no delayed prescription, CRP not repeated at re-consultation	1.00	Beta (1, 0)	[26]
**Resource Use Proportions**			
General Practitioner (GP) face-to-face consultation for lower respiratory tract infection (LRTI)	0.95	Beta (20, 1)	Raw data [26]
GP face-to-face consultation for acute respiratory tract infection (ARTI)	0.99	Beta (77, 1)	Raw data [26]
Independent Nurse Prescriber (INP) face-to-face consultations for LRTI	0.05	Beta (1, 20)	Raw data [26]
GP face-to-face consultation for ARTI	0.01	Beta (1, 77)	Raw data [26]
Telephone triage ^5^	0.01	Beta (1, 77)	Raw data [26]
**Costs (per unit)**			
GP consultation (9.22 minutes)	£38.00	Fixed	[39]
INP consultation (15 minutes consultation with band 7)	£13.25	Fixed	[39]
Telephone triage GP led (per telephone call)	£14.60	Fixed	[39]
Telephone triage nurse led (per telephone call)	£6.10	Fixed	[39]
Point-of-care (POC) CRP testing (per test)	£9.58	Fixed	Table 4
Amoxicillin capsules (500 mg three times daily for 5 days)	£0.91	Fixed	[40]
Amoxicillin capsules (500 mg three times daily for 7 days)	£1.27	Fixed	[40]
Clarithromycin tablets (500 mg twice daily for 7 days)	£2.23	Fixed	[40]
Dispensing rate for community pharmacists (per item)	£1.62	Fixed	[41]
Emergency ambulance ^3^ (per adverse drug reaction)	£236.00	Fixed	[43]
Emergency medicine ^4^ (per adverse drug reaction)	£362.00	Fixed	[43]
**Utilities**			
Utility (EQ-5D-3L score) ^6^			
U.K. population norm age 45–54 years	0.8470	1-Gamma (1.0000, 0.0015)	[44]
LRTI	0.6750	1-Gamma (1.0000, 0.0033)	[45]
Upper respiratory tract infection (URTI).	0.7970	1-Gamma (1.0000, 0.0020)	[44,46,47]
Anaphylaxis (adverse drug reaction) weight	0.5	Fixed	[47]
**Symptom Duration (days)**	**Median**		
Patient reported time to full recovery: LRTI CRP	15.5	Beta (2.8, 5.5)	[18]
Patient reported time to full recovery: LRTI Standard care	20	Beta (4.4, 4.5)	[18]
Patient reported time to full recovery: URTI CRP	14	Beta (2.3, 6.2)	[18]
Patient reported time to full recovery: URTI Standard care	14	Beta (2.0, 7.0)	[18]

^1^ Distribution used in probabilistic sensitivity analysis ^2^ Standard care for symptoms of ARTI for >12 hours where the antibiotic decision is unclear. ^3^ National average unit cost for Ambulance ASS02 See and treat and convey. ^4^ National average unit cost for VB01Z Emergency Medicine, Any Investigation with Category 5 Treatment. ^5^ Telephone triage lead ratio assumed to be equal to face-to-face ratio in the LRTI according to protocol model. ^6^ Utilities for 365 days have been adjusted in the 28-day model.

**Table 4 antibiotics-07-00106-t004:** Cost of point-of-care (POC) C-reactive protein (CRP) testing according to the CRP POC Testing Guidelines for Wales

Item	Cost (£)	*n*	£ Per Test	References / Assumptions
**Resource Use**				
Number of tests per GP practice	-	376	-	Projected from Hughes (2016) [26] assuming constant rate of uptake
Estimated life of the CRP Analyser (years)	-	5	-	Manufacturer quote (Alere)
**Fixed Costs**				
Afinion CRP analyser	1500.00	-	-	Alere Afinion AS100 analyser (Alere; MA, United States)
Connectivity	120.00	-	-	Betsi Cadwaladr University Health Board (BCUHB) estimate
Printer	250.00	-	-	Equal life to analyser
Scanner	125.00	-	-	Equal life to analyser
Total analyser set-up cost	1995.00	-	1.06	Calculated using machine life and number of tests per year
**Annual Costs**				
Associated connectivity cost	20.00	-	0.05	BCUHB estimate
Internal quality control (IQC)	136.00	-	0.36	Guidelines for Wales [48]
External quality assurance (EQA): Wales External Quality Assessment Service (WEQAS)	240.00	-	0.64	Guidelines for Wales [48]
Laboratory support (including travel, training, competency, clinical interpretation, quality, and troubleshooting support)	468.92	-	1.25	BCUHB estimate based on mid-point of AFC scale 2017 at each band and 28.1% on costs
Maintenance cost (annual after 3-years)	280	-	0.30	3-year warrantee
Total annual support costs	£976.92	-	2.60	
**Variable costs**				
Cartridge/reagent (per test)	-	-	3.50	
Health care assistant (HCA) time	-	-	2.42	Band 4 for 5 minutes [39]
Total variable costs	-	-	£5.92	
**Total cost**				
Total cost per test	-	-	£9.58	

^1^ Price year 2016–17.

**Table 5 antibiotics-07-00106-t005:** Model parameters for sensitivity analysis.

No.	Parameter	Point Estimate	Distribution ^1^	Assumptions/References
	Lower respiratory tract infection (LRTI) per protocol Probabilities			
	C-reactive protein (CRP) < 20 mg/L	0.70	Dirichlet (14, 6, 0)	Raw data [26]
	CRP 20–100 mg/L	0.30	Dirichlet (6, 14, 0)	Raw data [26]
	No antibiotics at index consultation | CRP < 20 mg/L	1.00	Dirichlet (14, 0, 0)	[11], Raw data [26]
	Delayed prescription at index consultation | CRP 20–100 mg/L	1.00	Dirichlet (6, 0, 0)	[11], Raw data [26]
	Delayed prescription not dispensed| CRP 20–100 mg/L	1.00	Beta (1, 0)	[26]
	No re-consultation within 28 days | CRP < 20 mg/L	0.93	Beta (13, 1)	Raw data [26] LRTI re-consultation with CRP < 20 mg/L
	Re-consultation within 28 days | CRP 20–100 mg/L	0.00	Beta (0, 1)	Raw data [26] LRTI re-consultation with CRP 20–100 mg/L
	Repeat CRP at re-consultation| CRP < 20 mg/L	1.00	Beta (1, 0)	Assumption: CRP repeated at re-consultation if used at index consultation.
	No antibiotics at re-consultation | CRP < 20 mg/L	1.00	Beta (0, 1)	Assumption: antibiotics only indicated at CRP > 100 mg/L; Hughes et al. (2016) [26] reports no evidence of CRP > 100 mg/L
i	Probability acute respiratory tract infection (ARTI) re-consultation standard care = CRP study data	0.1143		[26]
	LRTI re-consultation standard care = CRP pilot study	0.0500		Raw data [26]
ii	Cost of antimicrobial resistance per prescription over 28 days^2^			
a	European	£0.15		[27]
a	U.S.	£4.77		[27]
b	Global	£17.83		[27]
iii	Cost of dispensing Item dispensing fee for dispensing doctor	£1.90		[41]
iv	Cost and probability of hospital admission Cost of hospital admission for presenting complaint^3^	£826.69		[43]
	Probability of hospital admission: Point-of-care (POC) CRP test	0.0088		[11]
	Probability of hospital admission: standard care	0.0035		[11]
v	Probability of antibiotic use in standard care			
	Antibiotic prescribing: standard care	0.53		[11]
vi	Cost of amoxicillin prescription			
	Amoxicillin capsules: 500 mg three times daily for 5 days	£0.91		[40]
	Amoxicillin capsules: 500 mg three times daily for 7 days	£1.27		[40]
vii	CRP analyser machine life 10-years	10-years		Assumption

^1^ Distribution used in probabilistic sensitivity analysis ^2^ Currency conversion and inflation calculations applied. ^3^ Unit cost for on total Healthcare Resource Group (HRG) activity (excluding excess bed days) DZ22Q Unspecified Acute Lower Respiratory Infection, without Interventions, with CC Score 0-4.

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
