# Peer review of "Cost-Effectiveness Analysis of the Use of Point-of-Care C-Reactive Protein Testing to Reduce Antibiotic Prescribing in Primary Care"

_antibiotics, 2018, doi:10.3390/antibiotics7040106_

Round 1
Reviewer 1 Report
The aim of this study is to estimate the cost-effectiveness of point of care CRP testing versus standard care in adults presenting with acute respiratory tract infections in primary care. This is a well conducted study and the methods used are appropriate. I have a few comments.
Given the fact that this is a model based economic evaluation, it would have been interesting for the authors to have considered the long term cost-effectiveness of POC CRP testing. Although, this has been stated as a limitation of the study, considering the long-term cost-effectiveness in relation to antibiotic resistance is a question that needs answering.
This study seems to infer that POC CRP is borderline cost-effective (less likely to be cost-effective for upper respiratory tract infections). However, base case results did not account for the cost of antibiotic resistance. A number of recent studies have concluded that economic evaluations that it is important to include the cost of antibiotic resistance in such studies. I therefore suggest that the base case analysis should include the cost of antibiotic resistance or there should be a justification as to why the base case analysis has not considered these costs.
The aim of using POC CRP testing is to help reduce inappropriate prescribing of antibiotics which in turn would lead to reduced antibiotic resistance in the long-run. Therefore the study should present a more detailed discussion of the results obtained and the implication for the development of antibiotic resistance.
Please check the first sentence of the conclusion (Section 3.7) and correct any grammatical errors: “POC CRP testing for adults with where the antibiotic decision is unclear”
What was the rationale for assuming a 5-year lifespan for the machine?
Author Response
Point 1: The aim of this study is to estimate the cost-effectiveness of point of care CRP testing versus standard care in adults presenting with acute respiratory tract infections in primary care. This is a well conducted study and the methods used are appropriate. I have a few comments.
Response 1: Thank-you for reviewing this manuscript and for your valuable comments.
Point 2: Given the fact that this is a model based economic evaluation, it would have been interesting for the authors to have considered the long term cost-effectiveness of POC CRP testing. Although, this has been stated as a limitation of the study, considering the long-term cost-effectiveness in relation to antibiotic resistance is a question that needs answering.
Response 2: We agree that the long-term consequences of antimicrobial resistance is an important issue. We considered the cost of antimicrobial resistance in the sensitivity analysis, using the three values for the cost of resistance reported by Oppong and colleagues [1]. Estimates were based on the annual cost of resistance in the United States (US) $55 billion[2], the cost of multidrug resistance in the European Union (EU) 1.5 billion Euro [3], and the cost of global resistance over a 35-year period $2.8 trillion annually [4]. The cost per prescription in each scenario was calculated as the cost of annual cost of resistance, divided by the annual number of prescriptions in each region – assuming antibiotic prescribing is the main cause of resistance. In our analyses currencies were converted into pounds sterling and then inflated to price year 2016/17. Given the uncertainties in these estimates (the authors of these reports acknowledge that the cost implications of antibiotic resistance is subject to considerable uncertainty), we consider that it would be less appropriate to have used this in the base-case analysis. However, we have now provided more details of the costs of antimicrobial resistance used in our analysis in the manuscript (lines 460-467) and extended the introduction (lines 72-74) and discussion (lines 239-267) of the issues associated with incorporating antimicrobial resistance into analysis (also see below).
References:
[1] Oppong R, Smith RD, Little P, Verheij T, Butler CC, Goossens H, Coenen S, Moore M, Coast J. Cost effectiveness of amoxicillin for lower respiratory tract infections in primary care: an economic evaluation accounting for the cost of antimicrobial resistance. Br J Gen Pract, 2016, 21.[2] Centers for Disease Control and Prevention. Antimicrobial resistance: no action today, no cure tomorrow. World Health Day: Media Fact Sheet. 7 Apr 2011. Available at: http://www.cdc.gov/media/releases/2011/f0407 _antimicrobialresistance.pdf (accessed 19/11/2018)
[3] European Centre for Disease Prevention and Control, European Medicines
Agency. The bacterial challenge: time to react. Joint technical report. Stockholm:
ECDPC, 2009. Available at: https://ecdc.europa.eu/sites/portal/files/media/en/publications/Publications/0909_TER_The_Bacterial_Challenge_Time_to_React.pdf (accessed 19/11/2018).
[4] O'Neill J, Tackling drug-resistant infections globally: final report and recommendations. 2016. Available online: https://amr-review.org/sites/default/files/160525_Final%20paper_with%20cover.pdf (accessed 26 June 2018)
Point 3: This study seems to infer that POC CRP is borderline cost-effective (less likely to be cost-effective for upper respiratory tract infections). However, base case results did not account for the cost of antibiotic resistance. A number of recent studies have concluded that economic evaluations that it is important to include the cost of antibiotic resistance in such
studies. I therefore suggest that the base case analysis should include the cost of antibiotic resistance or there should be a justification as to why the base case analysis has not considered these costs.
Response 3: We agree that antimicrobial resistance should be considered in economic evaluations of interventions to reduce antimicrobial resistance. Unfortunately, however, as mentioned above, we were limited by the data available. Inclusion of costs and utilities related to antimicrobial resistance need to consider the context of resistance, uniform resistance patterns vary by context, bacteria specified and drug type. There are societal implications to consider which are beyond the scope of the analysis to quantify reliably. Indeed, there are no reliable estimates in the literature, meaning that we were very restricted in our ability to model the impact of resistance on cost-effectiveness. Because of this, we took the decision to include these broader estimates only in sensitivity analysis. A recent report by the Policy Research Unit in Economic Evaluation of Health & Care Interventions, in York [1], highlights the significant challenges of considering antimicrobial resistance in economic evaluations. We have extended our discussion on future research directions to include the need to address challenges, such as uncertainties in data and the future, indefinite time horizons, capturing the health outcomes and costs incurred by the wider population (lines 239-267).
References:
[1] Rothery C, Woods B, Schmitt L, Claxton K, Palmer S, Sculpher M. Framework for value assessment of new antimicrobials. Policy Research Unit in Economic Evaluations of Health & Care Interventions, York, September 2018. Available online: http://www.eepru.org.uk/wp-content/uploads/2017/11/eepru-report-amr-oct-2018-059.pdf (accessed 19/11/2018).
Point 4: The aim of using POC CRP testing is to help reduce inappropriate prescribing of antibiotics which in turn would lead to reduced antibiotic resistance in the long-run. Therefore the study should present a more detailed discussion of the results obtained and the implication for the development of antibiotic resistance.
Response 4: We have edited the discussion to present a more detailed discussion on the observed reduction in antibiotic prescribing (lines 170-173) and the value / wider implications of reducing antimicrobial resistance (lines 239- 267).
Point 5: Please check the first sentence of the conclusion (Section 3.7) and correct any grammatical errors: “POC CRP testing for adults with where the antibiotic decision is unclear”
Response 5: Typographical error corrected.
Point 6: What was the rationale for assuming a 5-year lifespan for the machine?
Response 6: The 5-year lifespan of the machine was based on the manufacturers estimate. The manuscript has been revised to include this detail (line 366) and a sensitivity analysis of 10-year life span has been included. The results (line 150 & Table 2) and methods (line 459 & Table 5) have been updated accordingly.

Reviewer 2 Report
Title: Cost-effectiveness analysis of the use of point of care C-reactive protein testing to reduce antibiotic prescribing in primary care
General comments:
The authors have undertaken an assessment of cost and patient outcomes related to antibiotic prescribing in the context of management of respiratory tract infection using point of care c-reactive protein testing as a metric of inappropriate prescribing. This is an important topic and likely to be of high interest among a broad range of clinical care providers, especially those practicing in infectious disease, as well as stake holders in health care administration. Minor issues with grammar need to be addressed throughout and occasional problems with vague wording or formatting.
Specific Comments:
Abstract:
Line 22: This sentence is confusing as written. The word ‘both’ is used in a sentence with three descriptors.
Line 23: The connection and what is meant between ‘sensitivity’ and ‘scenario’ analyses and ‘characterize uncertainty’ could be more precise. (As per sections 4.2.4 and 4.2.5)
Line 25: There is inconsistency in the use of cost-utility/effectiveness
Line 27: Need to define LRTI
Methods:
Line 307: Rephrase or spell out to avoid starting a sentence with a number.
Author Response
Point 1: The authors have undertaken an assessment of cost and patient outcomes related to antibiotic prescribing in the context of management of respiratory tract infection using point of care c-reactive protein testing as a metric of inappropriate prescribing. This is an important topic and likely to be of high interest among a broad range of clinical care providers, especially those practicing in infectious disease, as well as stake holders in health care administration.
Response 1: Thank-you for reviewing this manuscript and for your valuable comments.
Point 2: Minor issues with grammar need to be addressed throughout and occasional problems with vague wording or formatting.
Response 2: We have reviewed and edited the manuscript accordingly.
Point 3: Line 22: This sentence is confusing as written. The word ‘both’ is used in a sentence with three descriptors.
Response 3: We have edited this sentence to read: “Analyses considered (i) pragmatic use of testing, reflective of routine clinical practice, and (ii) testing according to clinical guideline.”
Point 4: Line 23: The connection and what is meant between ‘sensitivity’ and ‘scenario’ analyses and ‘characterize uncertainty’ could be more precise. (As per sections 4.2.4 and 4.2.5)
Response 4: We have edited this sentence to read: “Threshold and scenario analysis were performed to identify cost-effective scenarios.”
Point 5: Line 25: There is inconsistency in the use of cost-utility/effectiveness
Response 5: The term “utility” has been removed to avoid confusion.
Point 6: Line 27: Need to define LRTI
Response 6: We have edited the first use of LRTI to “lower respiratory tract infections (LRTI)”
Point 7: Line 307: Rephrase or spell out to avoid starting a sentence with a number.
Response 7: We have edited this sentence (now line 347): “Local prescribing data showed that in this primary care cluster, 22% of amoxicillin prescriptions (all indications) were prescribed for 5-days.”

Reviewer 3 Report
It is not clear how the study was conducted.
The non-expert reader may not understand what they are (in the sense of the type of patients) and there are patients or if there are references to previous pilot studies.
We talk about a model, what are the data on which the model was created. Furthermore, why is the people described only in the results?
There is no appropriate context. I would suggest the authors greater clarity in the description of the study. Surely the introduction must be improved
Author Response
Point 1: It is not clear how the study was conducted.
Response 1: Many thanks for taking the time to review this manuscript. We have revised the manuscript to clarify that this was a model based on empirical data (lines 79-80).
Point 2: The non-expert reader may not understand what they are (in the sense of the type of patients) and there are patients or if there are references to previous pilot studies.
Response 2: The economic evaluation is a model based on empirical data from a published study. The published study is described in lines 310-321. The model population and treatment pathways are described in section 4 (e.g. lines 281-283; 309-333).
Point 3: We talk about a model, what are the data on which the model was created. Furthermore, why is the people described only in the results?
Response 3: All parameters and sources of data are detailed in table 3.
Point 4: There is no appropriate context. I would suggest the authors greater clarity in the description of the study. Surely the introduction must be improved
Response 4: The study is reported according to the CHEERS guideline [2]. The introduction, and manuscript have been revised to reflect the comments of all three reviewers.
References:
[2] Husereau D, Drummond M, Petrou S; ISPOR Health Economic Evaluation Publication Guidelines-CHEERS Good Reporting Practices Task Force. Consolidated Health Economic Evaluation Reporting Standards (CHEERS)–explanation and elaboration: a report of the ISPOR Health Economic Evaluation Publication Guidelines Good Reporting Practices Task Force. Value Health, 2013, 16, 231–250.

Round 2
Reviewer 3 Report
No comments